# Co-Creation in Health 4.0 as a New Solution for a New Era

**DOI:** 10.3390/healthcare11030363

**Published:** 2023-01-27

**Authors:** Norbert Laurisz, Marek Ćwiklicki, Michał Żabiński, Rossella Canestrino, Pierpaolo Magliocca

**Affiliations:** 1Department of Public Management, Cracow University of Economics; 31-510 Krakow, Poland; 2Department of Management and Quantitative Studies, Parthenope University of Naples, 80133 Naples, Italy; 3Department of Economics, University of Foggia, 71122 Foggia, Italy

**Keywords:** Health 4.0, co-creation, healthcare, efficiency of healthcare services, usability of healthcare solutions, e-health

## Abstract

Previous research on co-creation in healthcare indicates that the use of co-creation in the design process of health solutions influences their greater acceptance and adaptation, resulting in greater efficiency of health services and higher usability of implemented health solutions. Analysis of adaptation and acceptance of new technologies reveals the problem of misunderstanding and the need for more trust in modern tools implemented in the healthcare system. The remedy may be the use of co-creation in the process of developing modern medical products and services. This article’s main purpose is to explore the co-creation process in Health 4.0, which is understood as the development of healthcare through the application of methods and tools of the Fourth Industrial Revolution. The literature review provided insights for an analytical framework—the co-creation matrix. We analyzed the case of the Italian medical platform Paginemediche.it to reveal the actors’ engagement in co-creation. The results demonstrated different levels of engagement in improving the efficiency of implementing medical and technological solutions. Both theoretical and practical analysis proved that the co-creation matrix helps more precisely define the scale and scope of co-creation in Health 4.0.

## 1. Introduction

Co-creation in a healthcare system calls for establishing a sustainable partnership between healthcare professionals and patients [1]. Although this approach is still being explored, some studies show theoretical sophistication [2,3]. This paper extends this research strand into a new area: Health 4.0 [4]. We understand Health 4.0 as the development of healthcare within the new opportunities created by the Fourth Industrial Revolution, which has transformed the entire economy, including manufacturing and services, into a new paradigm based on smart and sustainable business [5]. This change is bringing significant improvements in productivity, quality, and customer satisfaction. At the same time, technological sophistication is becoming a significant barrier to disseminating new solutions and thus requires the implementation of measures to strengthen the adaptation and acceptance of these solutions by medical personnel and patients [6,7].

Technological changes have strongly affected the healthcare field. Health centers, hospitals and many other healthcare institutions are introducing various diagnostic and treatment options based on data from modern devices. The use of machine learning and artificial intelligence (AI) algorithms has increased significantly [8]. New technologies are creating new opportunities that not only change the way we treat, prevent, or diagnose, but significantly affect management processes and the way health services are delivered [9].

If designed well, these solutions can significantly improve the quality of healthcare, patient experience, and physician satisfaction [10]. Success in this area raises the efficiency of the creation process and maximizes the usability of products and services. We understand efficiency, broadly, as the positive effect of a given process, while we relate usability to the product and/or the service, its value, and the quality of its performance. As a result, we define usability, in its classical sense proposed by W. S. Javons, as the main attribute of the value of a given thing. This thought is developed by later studies that relate value to satisfaction and usability [11]. In the case of our analysis, the key aspect is the acceptance of the product in the market by patients and physicians, which we implicitly define as the maximization of utility from the consumption of a given product or therapy administered using a given product or service. We can achieve such maximization of utility precisely through co-creation, i.e., the broad involvement of a wide range of stakeholders in the entire development process, including especially end users—patients and physicians.

A key aspect in the design, implementation, and proper use of new health technology solutions is to leverage users’ experience and possible involvement (patients, physicians, etc.) at every stage of the development process. The participation of users/consumers in the development process is called co-creation, and its application significantly improves the quality, usability, and social acceptance of new solutions [3].

The main goal of our article is to explore the co-creation process in Health 4.0, which is understood as the development of healthcare through the application of methods and tools of the Fourth Industrial Revolution (FIR). For this purpose, we identify actors’ roles in co-creating Health 4.0 solutions and determine the occurrence of co-creation in the service development phase. We verify the scale and scope of co-creation using the co-creation matrix as an analytical framework. We apply this framework to a selected case study. 

We demonstrated that the actors’ engagement in the service development process is unevenly occurring with the dominant role of suppliers and the limited involvement of customers. The results demonstrated different levels of engagement in improving the efficiency of implementing medical and technological solutions. Both theoretical and practical analysis proved that the co-creation matrix helps more precisely define the scale and scope of co-creation in Health 4.0. We argue that a detailed framework is required for future research and point out the areas for detailed and in-depth study.

Our paper is organized as follows. First, we present the characteristics of Health 4.0 and the solutions associated with this idea. Then, we explain co-creation in health services. We discuss the phases of service management and the actors involved in the process. After describing the theoretical background, we present the framework to assess the scale and scope of co-creation. We selected the case based on a systematic literature review—a solution created by the tech start-up Paginemediche. Next, we confront the obtained results with posed questions and provide theoretical and practical implications.

## 2. Co-Creation in Health 4.0

### 2.1. Health 4.0 Solutions

The main components of Industry 4.0 are the Internet of Things (IoT) and Artificial Intelligence (AI). The development of new technologies combined with the omnipresent Internet moves the economy at all levels and aspects toward increased digitalization, automation, and integrated control systems [12,13,14]. The catalog of new types of technologies, new inventions, and related applications is constantly expanding. There are collaborative robotics, industrial IoT, data analytics, additive manufacturing/3D printing, and big data analytics algorithms, as well as eServices, which is now one of the most visible examples of new technology available in the healthcare market [15,16,17]. From the healthcare perspective [18], we have new robots in surgery [19], new implants and prosthetics including 3D printed ones [20,21,22], new ways of gathering and analyzing medical data [23], new medical procedures, devices that allow the constant monitoring of the health indicators of their users or track real-time health data when a patient self-administers a therapy, and devices that auto-administer therapies for patients [22,23]. 

These new medical technologies may be divided into market segments as part of the embedded Fourth Industrial Revolution in production processes, products, and services [9]. Industry 4.0, from the healthcare perspective. may be perceived as solutions for individuals (e.g., broadly defined eServices, such as wearable technology, the Internet of Medical Things) [24,25], solutions for healthcare institutions (e.g., bigdata for clinical trials) [26,27], and new medical treatment and health therapies (new medical procedures) [23,28]. In this paper, we define Health 4.0 as healthcare development under the new possibilities created by the Fourth Industrial Revolution, which is also called Industry 4.0 [12]. It enhances technology-enabled care (TEC) [29].

### 2.2. Co-Creation in Healthcare Services

Consumers used to be treated as passive stakeholders. Today, business and science create new opportunities for stakeholder impact on value creation [30,31]. In this way, the construct of the value chain is broadened, which is why researchers currently study value networks and the multi-directional impact of many stakeholders on the value creation process [32]. Co-production—understood as multiple actors engaging in value coproduction activities—is becoming crucial in that process, which must be distinguished from the co-production of stakeholders [33]. It is defined as value co-creation, and the most important element that distinguishes classic product creation in the value chain from value co-creation is consumer involvement in that process [32,33]. The company is the designer and supplier of the product per se, but the value is created in interaction with the consumer [1,34,35]. The consumer becomes a co-creator, creating value during the in-use experience [36,37]. This means value is created due to a co-creation process, in which the most important relationship is between a provider and a customer [38]. Today, enterprises widely involve consumers in the process of value cooperation creation and product development [39]. 

Increasing the participation of consumers at all stages of the implementation of projects in Health 4.0 strengthens their involvement in the process of creating or testing those technologies [39], which increases the efficiency of the whole process. From that perspective, dialogue, access, and cooperation are most important in terms of consumer participation in the creation process. They are pre-defined key building blocks of consumer–company interaction underlying value co-creation [33]. Consequently, the sustainability of the Health 4.0 creation system depends on the network of stakeholders involved in the process—mainly end users [40]—as well as on the infrastructure that facilitates communication between them and the enterprises that create and implement products and services in this area [41,42].

A review of the literature on the implementation of new technologies in the health and healthcare system indicates that the complexity of these technologies slows the creation of new products and hinders their implementation process [1,43,44]. The research also shows that production and implementation require special knowledge, experience, and extensive cooperation with patients in the process of creating and carrying out projects [45]. To sum up, broad cooperation with stakeholders—mainly with patients—is a pre-condition for successfully creating and implementing Health 4.0 products/services. An important element of the development of Health 4.0 is the extensive cooperation of suppliers with consumers and other stakeholders during the process of adaptation to new technologies. The tool that allows suppliers to successfully implement the process of creating value in Health 4.0 is co-creation [34,38,46,47,48,49,50,51].

### 2.3. Co-Creation and Service Management Phases 

The process of creating a product or service as a result of creating a new value is carried out according to established patterns [52]. Stakeholder involvement is important at many stages, especially for consumers [33,36,42,43,45,49]. From this perspective, the description of the process of creating new products and consumer involvement must include consideration of the specificity of Health 4.0. In the case of Health 4.0, the authors particularly emphasize the key role of the implementation phase [41,50,51,52]. 

Popular classification of the project life cycle divides the process as follows: (1) initiation, (2) planning, (3) implementation, and (4) closing [53]. Some authors complete that classification with a phase of long-term duration (e.g., a monitoring and control phase) that is crucial in terms of development or emphasize the aspect of a feasibility study [50,54,55,56,57,58]. From the perspective of Health 4.0, the classification used for implementing information systems corresponds to the systems-development life cycle. That classification is more detailed and, in effect, it is possible to analyze products, services, and technologies. It contains (1) planning, (2) analysis, (3) design, (4) implementation/delivery, and (5) maintenance [3,8,38,59].

Based on the literature review, we resolved that in the case of the co-creation analysis, we would use the modified classification of life-cycle development. Our framework’s service management phases constituted planning, design, delivery, and maintenance. We also found that analysis was not a separate phase, but an integral part of each of the phases of the creation process proposed by Zhang [60]. Therefore, we abandoned the analysis phase in this classification. 

Planning: The important elements of this phase include initiation, a feasibility study, risk, strategy, a schedule, and a budget. To formulate the appropriate implementation strategy, one of the most important steps is to identify barriers and facilitators in the process [48,51]. Understanding the determinants of patients’ behavior is necessary to successfully implement Health 4.0 projects [54]. The research shows the need for cooperation in each project implementation phase, especially during the planning phase [56]. The participation of stakeholders during the planning phase promotes the development of projects, their implementation, and their stability [61]. 

Increasing consumer participation in this phase provides practical and conceptual help [41,61]. The patient’s (consumer’s) involvement in the planning phase allows a change of vision and adjustment of the implemented project to the needs and requirements of future users [28,51,60,62].

Design: This phase addresses the problems associated with creating a product, a system, and a technology [32,48]. Designing facilitates the implementation of all elements specified in the planning phase [47,63]. From the point of view of creating the system, the design phase means enriching the planned product, system, and technology with many details, especially technical ones [64,65].

In this phase, co-creation allows designers to confront the realities of customer needs and expectations and barriers to practical usage [28,32,39,60,61,62]. It leads to better decision-making during the phase and changes the customer–supplier/provider relationship [66]. Due to this, the design is often more efficient [50]. Based on such experience, it is possible to design the target approach, needs, and points of view [34]. 

Delivery/Implementation: This is the most challenging phase; many companies need help with completing this phase [48,57]. This phase’s key element is assessing test results and planning further procedures [67]. In this phase, cooperation with doctors and hospitals is crucial because, based on their opinions, the demand for medical services is created [68]. From that perspective, earlier cooperation increases the chances of more effective activities in the implementation phase [45]. The effectiveness of this phase depends, inter alia, on a partnership during the planning phase among all stakeholders [69].

A specific feature of this phase is the high cost of its implementation. Implementing technologies, products, and, especially, entire Health 4.0 systems is particularly expensive [2]. Many projects proceed as test projects, and most remain in a permanent pilot phase [39,52,61]. This means that several Health 4.0 innovations cannot proceed from a pilot phase to actual implementation [61]. Insufficient cooperation with doctors, hospitals, and end users is the reason for this inertia [45,56]. From that perspective, active cooperation with external stakeholders will facilitate a reduction in the costs of the testing phase and better preparation of the product/service for progress through the testing phase toward the final implementation phase [56]. 

Maintenance: During this phase, suppliers must support the operational effectiveness of the product/service. This phase requires constant monitoring, corrections, updates, and troubleshooting [52]. Complexity, the strength of which increases with the duration of the project, is critical in this phase. That complexity enforces broad and continuous cooperation among the stakeholders. The lack of comprehensive cooperation creates a gap between the implementation and maintenance phases [70]. As a result, the process is less sustainable and effective, particularly in Health 4.0 technology [62]. That, in turn, results in lower adaptability and durability of products/services. 

Patients interact with, adopt, and use technologies, in great measure, because they perceive that the technologies may be used for achieving desired goals [1,38,63]. Therefore, cooperation with providers and patients to identify those goals is essential for the successful and sustainable maintenance of Health 4.0 solutions [71]. Such cooperation allows the product or service to be constantly updated and adjusted to the needs and expectations of providers and patients [62]. This shows that greater involvement of patients in decision-making, evaluating, testing, and designing in each phase of creation and implementation should be a priority [61].

### 2.4. Co-Creation and Actors 

Stakeholders in the health care system can be identified as doctors, nurses, trade unions, insurers/payers, pharmaceutical and medical device manufacturers and patients [41]. The contemporary approach toward the definition of the healthcare system proposed by the World Health Organization (WHO) expands the actors to include leaders associated with governance and healthcare human resources [72]. In consequence, two types of actors appear: service providers and service suppliers. In any event, actors may be classified as regulators, providers, payers, suppliers, and patients [73]. Below, each of those groups is described.

Regulators: Ministries of health and those persons who oversee health issues at the regional and local levels are regulators. They set system boundaries and define the rules according to which actors perform. It is plausible to state that regulators represent the initial mode of governance and oversee the whole system. They have legal tools to evoke change [41,64,66]. This means that their actions affect the entire population. The healthcare sector is a special example of a comprehensive system with high social sensitivity [49,50]. It is impossible to create a sustainable healthcare system without stakeholders’ active participation and involvement in the decision-making process [55,74]. 

Regulators always operate at the macro level corresponding to the project, the product, or the actors. There is no direct involvement of regulators in the process of value creation. Therefore, we decided to highlight their importance and presence in the whole process and abandon the analysis of actors in micro cases, i.e., in the case study presented in this article. 

Providers: Providers are unusual actors in the healthcare system. On the one hand, they can be treated as the supply side of the system, because they provide, suggest, and convince patients to use specific products or services. On the other hand, together with the patients, they are consumers of those products, as well as end users [55,64,74]. This group represents human resources in the healthcare system and consists of doctors and nurses, as well as entities (organizations) such as hospitals or care homes. Therefore, we distinguish between individual and institutional service providers [41].

The active participation of providers in the process of creating a product/service determines quality, adaptability, and outcomes [29,51]. In the majority of cases, their role in the co-creation process is similar to that of patients. However, extensive and practical knowledge and experience are significant in facilitating a greater extent of cooperation [45]. Their lack of emotional involvement is an additional advantage [74]. 

Payers: The group of payers create statutory and private health insurance. However, due to the financial system, their role is significant in maintaining the system’s operations. In particular, statutory health insurance (SHI) appears to be a passive actor [64]. The strongly differentiated role of payers, depending on the healthcare system, is an important aspect in this case. Consequently, it is not easy to define a payer’s unambiguous role in the product-creation process [45]. 

A payer’s function has a narrow scope (i.e., it is limited to the financial aspect) and payers are not observed taking part in co-creation. However, healthcare reform encourages the co-creation of new insurance products and tailor-made tariffs, due to the increased knowledge of market-savvy consumers about SHI schemes [75].

Suppliers: The group of suppliers facilitates service provision. It embraces pharmacies, pharmaceutical companies, medical device companies, and ICT companies. Suppliers are most often the initiators of the whole process and the creators, beginning with the planning phase and continuing until successful implementation and achievement of deliverables. Their role is crucial, because they are the ones who initiate the process and decide on the scale of stakeholder involvement [55]. 

Within the framework of co-creation, the suppliers’ role resembles that of providers. They are not directly involved in patient relations; however, they offer the necessary equipment and technology. Within this context, the specificity of products/services offered as part of Health 4.0 brings about the need for cooperation with end users [6]. In terms of contemporary economics, suppliers engage stakeholders during each subsequent phase of product development [60]. Suppliers participate in every phase of the process.

Patients: Patients are the largest group in this collation, although they have the weakest influence on the system. The patient is the end user in the product creation process. From a market perspective, the patient creates the demand for Health 4.0 products. Therefore, the provider—the entity that co-creates that demand jointly with the patient—is a crucial element. The patient performs a functional assessment of the product, while the provider is responsible for the substantive aspect. Those two actors shape the demand for Health 4.0 products and services [56,60,64].

Nevertheless, the relevant literature regarding co-creation emphasizes the activation of patients who are consumers [7]. The requirement for “patient empowerment” results in a diversified extent of patient inclusion in service management. For example, R. Palumbo [3] distinguishes among patient enablement, patient activation, patient engagement, and patient involvement. To this end, patient empowerment may also refer to other actors; however, as the research indicates, the patient and patient participation are critical in most stages of the product-development process [52,53]. In the case of medical services, the role of hospital staff may change, depending on the purpose of the product or service. Doctors and hospital staff members become end users of medical services. This is particularly visible in the cases of patient databases, monitoring applications, and platforms used for identifying disease or therapy. In such cases, doctors, nurses or other medical staff become the consumers (end users), while the patient does not participate in the process at all. 

Nowadays, co-creation has become a way of creating and implementing many projects in the field of creating products/services and technology. The research indicates that Health 4.0 requires cooperation to the greatest possible extent. In this way, the likelihood of positive transitions from testing phases to implementation phases and, further, to long-term product maintenance increases significantly.

## 3. Materials and Methods

We complemented the theoretical and conceptual analysis, which was conducted on the basis of published studies on co-creation in both the area of health and in cases of modern technology, by presenting the analytical use of the co-creation matrix. This matrix allowed us to analyze the scope and scale of co-creation use in the case of Health 4.0 solutions. The theoretical analysis of the issue allowed us to adapt the matrix to the co-creation requirements in Health 4.0. The basic information was obtained from peer-reviewed scientific articles. However, to complete the case description, it was necessary to supplement these data with additional information as part of the analysis of co-creation. Therefore, to complete the co-creation matrix, it was necessary to review non-scientific literature, reports, and interviews with the developers of this solution or its users. The process of case selection, analyzed using the results of the literature review, is presented below.

The research design consisted of three main steps: (1) the identification of case studies of comprehensive Health 4. 0 and co-creation, (2) the selection of the most informative example, and (3) the assessment of the scope and scale of co-creation in Health 4.0 ventures.

### 3.1. The Identification of Already Investigated Cases

We employed a systematic literature review protocol to establish the number of previously investigated examples of co-creation and Health 4.0. We found only a few papers relating directly to Health 4.0; therefore, we extended the search string to e-Health. We used the keywords "co-creation", "Health 4.0," or "eHealth" in three databases—the Scopus, Web of Science, and PubMed databases—without limitation to period or paper type. We excluded records that did not correspond simultaneously to co-creation and Health 4.0/eHealth. Finally, we identified twenty-three papers. Figure 1 presents the flowchart for the SLR for co-creation in Health 4.0, including eHealth.

### 3.2. The Selection of the Most Informative Example

Each of the examples was checked for an explanation of the technology used, the actors engaged, and the co-creation scope within the product/service life cycle (planning, design, delivery, and maintenance). We chose the case where the co-creation phenomenon was described in the most detailed way and where the involvement of actors appeared on the most significant scale. Another article by the authors of this paper, entitled ’The scope of stakeholders’ involvement in Health4.0 adoption: the perspective of co-creation’, provides a broader discussion of the literature review and the cases identified. 

The selected papers focused on the following areas:-the process—phases of creating a solution (12 articles);-the entity—actors (16 articles);-results—products/services (12 articles);-health policy and implementation (9 articles);-solution and/or health system effectiveness (17 articles);-current knowledge synthesis (literature review or case presentations) (6 articles).

Within the last group, only five articles discussed issues related to the co-creation process, its phases, its actors, the relationships between them, and the product/service. In contrast, only three of the articles additionally related the implementation of a solution to its effectiveness or to the effectiveness of the health system. Of these three examples, one was selected to present the issue: Paginemediche (in English, “medical page”). It was first described by L. Lo Presti, M. Testa, V. Marino, and P. Singer in 2019 [45]. We have chosen this case because it is the most comprehensive example in the literature of the whole phenomenon of co-creation in Health 4.0. Table 1 contains basic information about Paginemediche.

### 3.3. The Assessment of the Scope and Scale of Co-creation in Health 4.0 Ventures

We searched for additional data about the selected case to enrich the description with information about the roles of different co-creation actors in service provision. The company’s website and its content in the Italian language delivered necessary data. We applied the healthcare co-creation framework proposed by Palumbo [3], which we explained in the literature-review section, as a pattern-matching analytic technique [78]. We analyzed the data using, as codes, the main categories of the co-creation matrix: actors, co-creation areas, and service-management processes. The co-creation matrix depicts interrelations among key co-creation dimensions (actors vs. phases; see Table 2). Two research questions guided our analysis: (1) What could be the role of actors in co-creating Health 4.0 solutions? (2) In which service-development phase is co-creation observed? The co-creation matrix for the digital health platform (Table 2) comprises the results classified according to the main categories (actors and activities). In more extensive research, this matrix enables a comparison of the implementation of Health 4.0 solutions via the co-creation approach. Additionally, we used a co-production evaluation model to classify the selected case, using the scope and scale of co-creation in Health 4.0 ventures (Figure 2). Both matrices were filled by assessing the evidence and matching the findings to the proper category.

## 4. Results

### 4.1. Description

Paginemediche was one of the first digital health platforms in Italy. Such platforms support the exchange of knowledge between doctors and patients, as well as between doctors and their colleagues [79]. Paginemediche is a multifaceted platform aimed at offering personalized health services to patients who interact with doctors for direct consultations or appointments. By the end of 2021, more than 15 million users had accessed Paginemediche’s website, and more than 140,000 professionals had registered with MediciOnline and used the platform’s services. More than one million individual patients used digital support every month. 

In addition to the ability to contact and connect specialists with patients, the platform offers a range of usability aimed at increasing patients’ medical and health awareness, accessing medical advice on preventive and therapeutic measures, and information about the treatments themselves. Among other things, the platform facilitates finding advice by discussing a disease or symptoms via multi-channel communication. To increase the audience’s understanding of the content, topics are presented in written form and in the form of video advice. To this end, Paginemediche uses multiple avenues to reach patients, the most obvious being the website, but also via a dedicated app (Visitami; in English, “Visit Me”), a channel on LinkedIn, YouTube, or Tik Tok. Increasing accessibility and using a multi-channel and multi-threaded way of communicating with patients has not only marketing goals. These activities are also means of meeting the needs of customers and of professionals, who reported the need for such a form of communication as part of ongoing behavioral monitoring and stakeholder and customer collaboration. 

As a result of ongoing cooperation with clients, the platform is constantly improving its offerings. An example of such an additional service was the introduction of a chatbot dealing with Covid-19 cases, an initiative that was carried out in cooperation with the Italian Ministry of Health. The coronavirus chatbot was also designed to assist doctors in dealing with suspected cases. Its main advantages were pre-selection based on symptoms and significantly speeding up the process of identifying possible coronavirus-infected cases. This online decision-support system and self-triage application gained interest as being useful for traditional health surveillance [77]. 

The initiative won the award for best innovation in 2019. A year later, the content—published in tips, mini-lectures, or experts’ answers to frequently asked questions—to a substantive review. As a result, at the end of 2020, Paginemediche became a certified provider of real and substantive content in the field of medicine and health (via HONcode certification). The certification significantly raised the popularity and recognition of Paginemediche. It has also resulted in the amount of content provided increasing significantly. Currently, the Paginemediche app and website are among the main sources of information on prevention, treatment, and how to proceed in health-related cases.

### 4.2. Actors 

The case describes doctors’ engagement with their peers and patients. On the one hand, patients design their own “personal area” by adding their biometric parameters and browsing preferences. Based on the individual’s profile, the so-called “digital footprint” is carried out to enable Paginemediche to collect data and content and coordinate services. As a result, patients have access to personalized health programs and digital therapies of interest, and they can interact with doctors for direct consultation or to make an appointment. Digital services such as “Expert Answers”, “Patient Kit”, and “Digital Therapy” are available to obtain a second medical opinion. Moreover, patients may consult apps and advice suggested by doctors in support of therapies, care, and prevention paths.

Doctors also have the chance to register themselves in a “professional area”, with medical-scientific insights, and collaborate with specialists (content providers) in the creation process for topics, while increasing their visibility and influencing the web through online doctors’ services. 

The use of doctors’ and patients’ opinions is the basis for applying modifications to the operation of the formal and substantive levels of the platform. The knowledge of doctors and medical professionals is used in all phases. In this case, the significant involvement of actors in the development and implementation process is evident. Significant sources of information are the patients, but their involvement in the creation process is one-sided. Although the use of knowledge and opinions is significant, and the acquisition of knowledge from patients takes place at many levels, the involvement of patients in the creation process is rather symbolic. Patients are treated as an essential resource of information.

A survey of bilateral interest (patients and physicians) is used to design directions or additions to data, advice, personnel, or services. While interviews confirm the emerging inclination to use artificial intelligence, to date the primary means of analysis are direct interviews with patients and doctors. 

### 4.3. Co-Creation Mechanisms/Areas

Co-creation occurs as part of service creation and delivery. Involvement in the planning and creation of the platform is particularly evident among doctors and medical professionals. In addition, doctors and patients consult on the quality and alignment of services with customer expectations during face-to-face meetings and consultations. In fact, informal patient–doctor interactions are the best sources of knowledge about the needs and opinions regarding the platform.

The emergence of coronavirus-derived COVID-19 emphasizes, much more than before, the need to improve health monitoring utilizing web support, as the Italian lockdown and the national healthcare system’s collapse (under the pressure of the pandemic spread) limited the access to healthcare services for citizens. In line with the mentioned arguments, Paginemediche provided a thematic chat allowing, on the one hand, healthy people to know how to protect themselves from the contagion and, on the other hand, infected patients to be monitored all the time. As Lo Presti et al. [45] noted, this type of platform strengthens stakeholder engagement by empowering its users: engagement and relationships are created among users—doctors and patients—as a result of providing solutions relevant to the needs of service recipients [79,80].

### 4.4. Co-Creation Matrix 

We discussed the results using the theoretical framework for depicting the co-creation occurrence in Health 4.0. The pattern-matching technique enables consistency and comparability of findings in future studies, enriching our knowledge about co-creation in Health 4.0 applications.

Based on Palumbo’s studies, we assumed that co-creation would occur in each phase of project implementation, as well as involving each actor’s participation in its implementation. The matrix (Table 2) indicates areas of co-creation’s appearance in Health 4.0. It also allows assessing the scale of co-creation activities in each phase and the involvement of a given group of actors. 

We observed that all actors were involved in the planning phase in this case. The situation changed in the design phase. Then, only suppliers and providers were the involved actors; therefore, they were considered as the main actors. Subsequent phases were characterized by much greater involvement of actors. However, in the case of consumers, it was evident that this group was treated as a subject of study and a data source, rather than as an active participant in the co-creation process. Interestingly, the payers appeared only in the planning phase; in the later stages, their involvement was limited to formal activities, control activities, and accounting activities (e.g., controlling expenses, budget execution, and making payments).

As part of the design phase, the suppliers were responsible for designing the system, by offering guidance and functionality to deliver accessible and valuable services [81]. In addition, they were a highly desirable consultant in each phase of the service-development process. Suppliers provided technological support and made recommendations on how to improve the health service. Among other things, their primary roles were to provide adaptable implementation and stability in system operation and to guarantee service availability. The requirement for suppliers to understand customers and their needs was critical for all phases [82]. 

The nature of the healthcare market shifts the responsibility to suppliers to evaluate the solutions offered. From this perspective, suppliers are the natural link between the developer and the ultimate consumer. In this context, they can be considered the "first consumer" in a complex consumption process. It is the doctor who first accepts the solution, product, or service, and then recommends the solution to the patient. In our example, the service offered was, on the one hand, medical in nature and, on the other hand, technical and digital in nature. Taking the latter into account, it is important to note the low scale of co-creation in the case of patients and their single-purpose involvement in the whole process. The lack of active participation of final consumers shows that the organization needs to fully use the tool of co-creation in the process of creating the offer and the service themselves.

Adapting the model used to evaluate co-production, allowed us to determine the type of co-creation. We classified Paginemediche as individual co-creation (co-production) (upper left in Figure 2), due to the in-depth activities of most actors, and. at the same time, the relatively narrow scope of their activities.

**Figure 2 healthcare-11-00363-f002:**
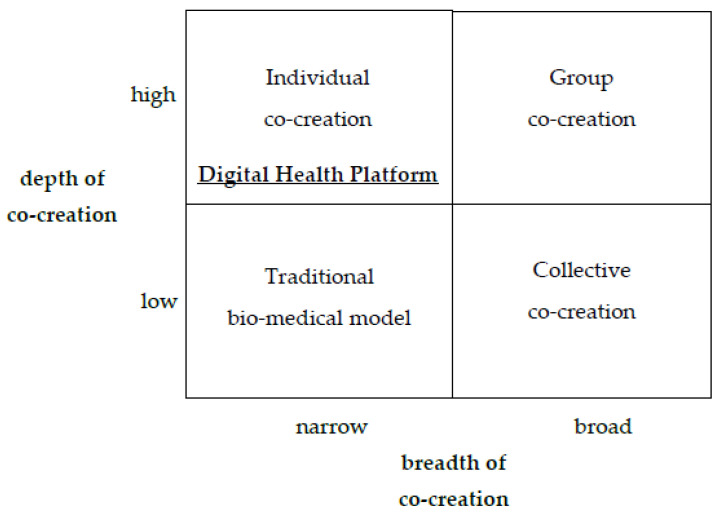
Co-creation Styles in Health 4.0 (an adaptation of Palumbo’s co-production styles in the healthcare environment matrix [3].

## 5. Conclusions

### 5.1. Theoretical Implications 

The answer to the question of the role and extent of involvement related to the role of actors in the process of co-creation of Health 4.0 solutions should be given on a theoretical level by discussing two issues. The first issue is the possibility of using co-creation in the creation process; in this case, the key aspects are the identification of the possible area of implementation of this process and the definition of a way/tool to measure the scope and scale of co-creation in the creation process. The identification of the area of realization of co-creation in the creation process and the involvement of actors in this process was carried out through a theoretical analysis of the co-creation process, especially co-creation in the area of health. As a result of this analysis, it became possible to propose a theoretical account of the areas, scale, and scope of co-creation in the process of creating solutions in Health 4.0. The final result was the development of a co-creation matrix and the proposed use of this tool to describe co-creation in Health 4.0, on both practical and theoretical levels.

The second and target issue considered in theoretical terms was the possible positive effects of such a process, which refers to the positive effects achieved. This issue is mainly related to the high levels of distrust and deficits in the understanding and ability required to use modern technologies. The issue of health and medical care is an extremely intimate area, and the introduction of new and unknown solutions often faces a psychological barrier for patients. Therefore, raising the efficiency of the creation process, which is understood as effectively combining the capabilities of all stakeholders (including end users, i.e., patients and doctors) in the creation process and maximizing the usability of products and services as a result of their best fit for the needs and expectations of patients and doctors, are key aspects. From this perspective, developing a solution with the involvement of the end user seems to be a favorable solution from both the medical and economic perspectives. As the Technology Acceptance Model explains, the perceived ease-of-use and the perceived behavioral control affect actual use [83]. The presented results show that the participation of patients and doctors as co-creators can take place in Health 4.0, but the scale and scope of co-creation depend on various factors, including system design and empowerment of participants in the process.

### 5.2. Practical Implications 

The answer to the second question, regarding the practical aspect of the appearance of co-creation in the service development process, directly refers to the results of the theoretical analysis, i.e., the use of co-creation to achieve better results in creating products and services in Health 4.0 and the analysis of the scale and scope of co-creation through the use of the co-creation matrix. The Paginemediche case allowed us to demonstrate the practical use of the co-creation matrix. The analysis carried out provided information on the scale and scope of co-creation in a specific case and showed how different the ways and the intensity of involvement of individual actors can be. 

In the analyzed case, the way of involvement of actors was classified on the basis of collected pieces of evidence, as an "individual co-creation" type. It showed that the co-creation process itself can proceed differently, depending on the specific case. This study demonstrates that the full scope of co-creation and the levels of patient empowerment are not evident in Health 4.0 applications. However, the issue undoubtedly warrants a broader analysis.

In summary, our work contributes to the existing theory of co-creation in the context of Health 4.0 solutions. The theoretical implications refer to the need for structured analysis of the co-creation phenomenon because of different levels of engagement of different actors in different phases. The general application of co-creation in Health 4.0 requires refinement. Co-creation is perceived mainly as involvement of customers [3,35,47,75,81], while we observed their limited engagement. The role of other stakeholders is less emphasized in the literature, while our study demonstrated the significance of third-party roles, i.e., that of suppliers. The practical implications resulting from Paginemediche are twofold. First, consumers’ engagement in service development was considered. The necessity to interact with future users is crucial not only for designing a product matching the needs of consumers, but also for the broader purpose of complementing the national health surveillance system [44,77]. Second, online engagement is necessary for delivering data for telemonitoring. This took place in Italy, a country that was strongly impacted by the Covid-19 pandemic, where barriers to engaging with new technology were weakened. We also pointed out the context of co-creation as a factor that is essential in the seeking of higher efficiency in medical products and services.

### 5.3. Study Limitations

The major limitation of this study was the difficulty in benchmarking more cases, due to the small number of academic articles analyzing this issue. Only 23 examples were identified in the systematic literature review, of which only five described in detail the co-creation process, the actors, and the product or service. We also reckon that the lack of description in all fields of the co-creation matrix may be due not only to non-existent relationships but also to an information gap. In response to these limitations, we propose to focus future research on collecting primary data to provide a more complete and richer picture of Health 4.0 solutions.

## Figures and Tables

**Figure 1 healthcare-11-00363-f001:**
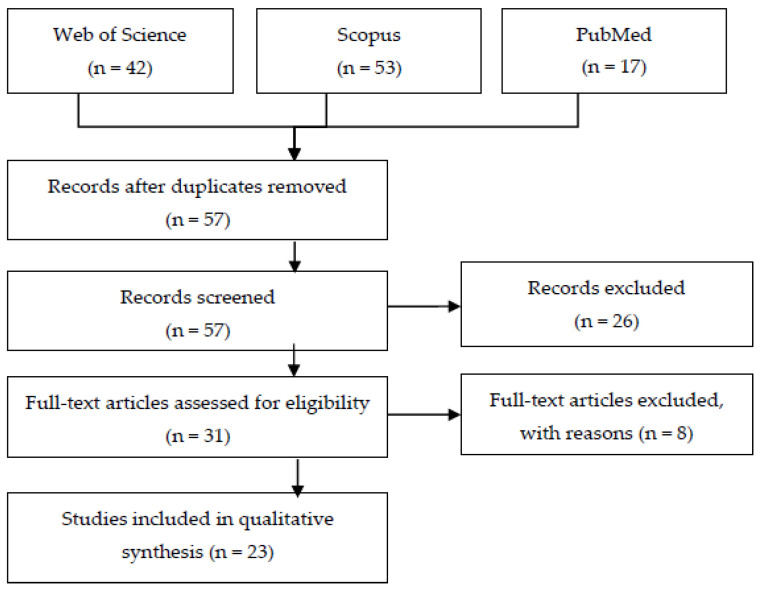
Steps in literature review on co-creation in Health 4.0 (including eHealth).

**Table 1 healthcare-11-00363-t001:** Synthesis of Paginemediche.

Name	Paginemediche
Type	Health engagement (internet communication) platform
Website	paginemediche.it
Country	Italy
Year of establishment	2015
Key areas	Communication with doctors–patients Online decision-support system Telemonitoring service for patients
Data sources	paginemediche.it; Lo Presti et al. (2019) [45]; Lo Presti et al. (2022) [76]; Tozzi et al. (2021) [77]

**Table 2 healthcare-11-00363-t002:** Co-creation matrix for digital health platform.

	Actors	Providers	Payers	Suppliers	Consumers
Phase					
**Planning**		Create solutions based on feedback from doctors and patients Formulate concepts for changing the content of the ways services are provided	Budgeting Controlling	Information support, real participation, and involvement in the planning process	Information support in the form of a passive and one-sided flow of information
**Design**		Healthcare service provides the design of the system	-	Define contents and medical topics as a result of the interaction with doctors—video visit service	-
**Delivery**		Analyze data Consult patients’ medical data Enrich own expertise	-	Provide contents and medical topics to be uploaded onto the platform (medical papers and scientific research)	Provide data into the system Share the data with members from their network
**Maintenance**		Create new solutions and digitally supported services	-	Build a knowledge asset Complement the content available online and take care of the quality of services offered Medical data analysis provides substantive support of the process of changes and updates in the services provided	Provide information using direct services and content and the services available online

Source: authors’ own elaboration.

## Data Availability

Not applicable.

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
