# Peer review of "Co-Creation in Health 4.0 as a New Solution for a New Era"

_healthcare, 2023, doi:10.3390/healthcare11030363_

Round 1
Reviewer 1 Report
Dear Authors,
thank you very much for your work. You picked a very interesting topic about which there is an undisputed gap in literature. Otherwise, you didn't make very evident if and how you filled that gap. In example, you started section 5 by claiming the contribution of your work, but actually your motivations sound more suitable to support the importance of co-creation in healthcare than to show if your work is of any help. Regarding that, you shall revise section 5 and hopefully introduction as well, in order to report in a substantiated way the contribution of your work from a scientific as well as practical point of view.
Besides, you shall define better the methodology you used by presenting in example a flow chart or something equivalent to give a clear picture of the methodology and you shall provide a detailed description as well.
Similarly, you shall explain better how you developed tables 2 and 3 and figure 2. They look as the real outcomes from your work, but I couldn't say now how you got there. You shall provide a better description of tools, data, everything needed to evaluate if the results are robust from a scientific point of view and the procedure you used is repeatable.
last thing, I wish you best luck for your work
Reviewer 2 Report
Thank you for giving me this opportunity to review the manuscript entitled, “ Co-creation in Health 4.0 as a new solution for a new era.”
The manuscript provides synthesized contents about Health 4.0 and co-creation of various actors. However, the outcomes should be clearly explained in detail. Some comments are below.
Abstract
In abstract, I am not sure about method and the data that could achieve the aim of this research. The purpose, method, data procedures and samples, brief results and implications should be reported.
Introduction
The purpose of this study is not clear.
On page 2, “The main goal of our article is to define a framework for analyzing the phenomenon of co-creation in the area of Health 4.0, which we have defined as the development of healthcare through the application of methods and tools of the Fourth Industrial Revolution.”
“the final step is to present how to use this tool using Paginemediche,it as an example.”
All the purposes are very ambiguous and the results should answer the goals. The definition or more explanation (Paginemediche,it) can be helpful for readers.
On page 5, “Live Cycle Development” would be one of the misspelling words.
Method
The specific explanation of this methodological approach should be explained. Is this a systematic literature review? Or meta analysis?
Selected samples
The specific information of articles (n=23) can be explained more in detail. Previous research often provide summary of the tables.
On page 9, Paginemediche,it and the website are mentioned again, but the data sources or the specific information about the data obtained from them.
results
Table 2 and figure 2 should be core contents of this study. Specific information about the outcomes of synthesized information can be useful for readers.
conclusions
Conclusions and implications are somewhat weak.
Typos and paragraphs
Overall, there are some misspelling and types in the manuscript. Please check them out and revise.
A sentence is a paragraph in the manuscript. For providing better information, please revise the manuscript.
Round 2
Reviewer 2 Report
Thank you for revision of the manuscript.
two of my biggest concerns is 1) not sufficient data and 2) a qualitative analysis approach or software.
1) I think a total of 23 articles would be too small. But when I closely look at the page 8. The results of this study seem to come from only three articles or only focused on three articles. This research provides interesting contents but the approach is not very scientific.
2) I am not sure about how authors organize and analyze the contents of the articles. Coding process, reliability and validity were not discussed.
Author Response
Dear Reviewer,
Thank you very much for your comments.
The number of articles identified -- 23 at total -- does not surprise us. We followed the PRISMA recommendation, which is commonly recognized as fulfilling criteria of methodological rigour, and we explained the decision made in qualifying papers for further analysis.
The number of papers can be perceived as too small, but we reason that it is because the combination of two recent concepts: health 4.0 and co-creation, is novel, and there is not much research about a joint phenomenon.
We have observed that even in papers describing it, some details are missing: thus, the scientific literature was insufficient. Therefore, to fill out the Co-Creation Matrix, we review non-peer-reviewed literature like reports or interviews with the solution's creators or its users. In order to present the Italian Digital Medical Platform as fully as possible, we used original sources that our team members analyzed.
The collection of these materials enabled us to fill the missing pieces of information and reach the saturation point, i.e., to create a sufficient picture of the studied phenomenon. Of course, some would say that it is still not fully satisfactory, and we pointed it out in the Study limitations:
The major limitation is the difficulty of benchmarking more cases due to the small number of academic articles analyzing this issue.
- We have clarified that we coded collected material using the main categories of the co-creation matrix as thematic codes: actors, co-creation areas, and service management processes.
Your comment allows us to better describe this element in our article. Our description, in this case, was not as precise as it should have been and could confuse the reader. Accordingly, the needed changes were made.
Best Regards
Authors